# Easy Anthropometric Measurements Are Representative of Baseline Values of Breast Q Values in Asymptomatic Women

**DOI:** 10.3390/healthcare12020268

**Published:** 2024-01-20

**Authors:** Giuseppe Catanuto, Nicola Rocco, Concetta G. Fichera, Ada Cinquerrui, Martina Rapisarda, Paolo Chiodini, Francesca Magnoni, Patrizia Dorangricchia, Valeria Sebri, Gabriella Pravettoni, Maurizio Bruno Nava, Francesco Caruso

**Affiliations:** 1Institution Humanitas Istituto Clinico Catanese, 95045 Misterbianco, Italy; giuseppe.catanuto@humanitascatania.it (G.C.); martinarapi96@gmail.com (M.R.); francesco.caruso@humanitascatania.it (F.C.); 2GReTA Fondazione ETS (Group for Reconstructive and Therapeutic Advancements), 80123 Naples, Italy; maurizio.nava@gmail.com; 3Department of Advanced Biomedical Sciences, University of Naples “Federico II”, 80131 Naples, Italy; 4Associazione Sant’Antonese Lotta ai Tumori, 95025 Aci Sant’Antonio, Italy; cg.fichera@gmail.com; 5Medical Statistics Unit, University of Campania “Luigi Vanvitelli”, 81100 Caserta, Italy; 6Division of Breast Surgery, European Institute of Oncology, IRCCS, 20141 Milan, Italy; francesca.magnoni@ieo.it; 7Applied Research Division for Cognitive and Psychological Science, IEO, European Institute of Oncology, IRCCS, 20141 Milan, Italyvaleria.sebri@ieo.it (V.S.); gabriella.pravettoni@ieo.it (G.P.); 8Department of Oncology and Hemato-Oncology, University of Milan, 20122 Milan, Italy

**Keywords:** breast cancer, breast measurements, quality of life

## Abstract

Background: Measurements of breast morphology are a determinant of the assessment of any surgical procedure, either reconstructive or cosmetic. This study aims to investigate the association between easy anthropometric measurements and values of quality of life assessed in a sample of asymptomatic women. Methodology: Healthy asymptomatic women were admitted for this study. The following measurements were assessed: height, weight, nipple to sternal notch distance, areola to infra-mammary fold distance (right vs. left), right–left nipple distance. The Breast Q questionnaire (Italian translation V.1, pre-op breast conservation surgery) in the following domains: satisfaction with breasts; psycho-social satisfaction; physical satisfaction; sexual satisfaction, which was used to assess breast-related quality of life. Results: One hundred and forty-five women responded to the breast Q questionnaire. The mean age of the sample was 44.3 years; the medium BMI was 24.1; Spearman correlation coefficients revealed that all the investigated values were negatively correlated to the “satisfaction with breasts” domain. Psychosexual satisfaction was associated with age; BMI; nipple to sternal notch distance. After normalization for age values, we observed that “satisfaction with breast” was, once again, highly correlated to BMI; nipple to sternal notch distance; areola to IMF distance. In all cases, the higher the values, the lower the scores. Conclusions: Distances between easy relevant anatomical landmarks are representative of patients’ breast-related quality of life in a population of asymptomatic women. These findings allow us to identify an ideal anthropometric framework that can be used as a validated surgical endpoint for cosmetic and oncological procedures.

## 1. Background

The female breast has a complex three-dimensional shape. Its morphology is determined by anatomical tissues retaining different viscoelastic properties. For this reason, it moves and warps, changing its shape under dynamic conditions [1,2].

Morphological estimates are a determinant of any reconstructive or cosmetic procedure [3]. Rigorous quantitative evaluation of volumes or surface curvature can be either misleading or not cost-effective depending on certain shapes [4], or complex technologies required to obtain reliable results [3,5].

Most of these measures cannot be taken routinely without altering the patient’s path, including at least a visit to the patients’ photography department.

For this reason, in this study, we aimed to investigate whether or not simple easy linear measurements taken during everyday practice can, in some ways, be representative of breast-related quality of life (BRQoL).

## 2. Materials and Methods

Healthy asymptomatic women undergoing annual clinical examinations as a prevention program of a charity institution (Associazione Santantonese per la Lotta ai Tumori- Aci Sant’Antonio, Catania, Italy) were admitted for this study. Before clinical examination, they were informed about the purpose of this study and signed a specific consent form. The assessment included an estimation of the following measurements: height, weight, nipple to sternal notch distance (right vs. left), areola to infra-mammary fold distance (right vs. left), and right–left nipple distance.

Breast ptosis was assessed in accordance with the Classification of Regnault. Women with grade 4 ptosis were excluded for the sake of uniform reporting of qualitative values.

After clinical examination, they were invited to fill in the Breast Q questionnaire (Italian translation V.1, pre-op breast conservation surgery) in the following domains: satisfaction with breasts; psycho-social satisfaction; physical satisfaction; sexual satisfaction. Questionnaires were collected anonymously, manually recorded, and stored into an electronic database under GDPR regulations.

### Statistical Analysis

Continuous variables were reported as mean and standard deviation (SD). Correlations between continuous variables were calculated with the Spearman correlation coefficient. The partial Spearman correlation coefficient was used to adjust estimates for age. A two-tailed *p* value < 0.05 was considered significant. Data were analyzed using SAS version 9.4 (SAS Inc., Cary, NC, USA).

## 3. Results

One hundred and forty-five women entered this study and responded to the Breast Q questionnaire. The mean age of the sample was 44.3 years; the medium height was 161 cm; the medium weight was 63.14, medium BMI was 24.1 (Table 1). The mean scores of the Breast Q questionnaire are reported in Table 2 (the MEANS procedure in the SAS output). Spearman correlation coefficients revealed that all the investigated values (age; BMI; nipple to sternal notch distances; distance between two nipples; areola to infra-mammary fold distance) were negatively correlated to the “satisfaction with breast” domain (Table 3). Psychosexual satisfaction was associated with age; BMI; nipple to sternal notch distance. The overall score was highly statistically associated with age; BMI; nipple to sternal notch distance; areola to infra-mammary fold distance. In all cases, Spearman correlation values were negative (Table 3). After normalization for age values, we observed that “satisfaction with breast” was, once again, highly correlated with BMI; nipple to sternal notch distance; areola to IMF distance. Overall satisfaction with breast irrespective of age was associated with BMI and nipple to sternal notch distance. In all cases, the higher the values, the lower the scores (Table 4).

## 4. Discussion

### 4.1. Breasts, Body Image and Quality of Life

Many studies underlined how breast perception could influence quality of life [6,7,8].

Breast perception is an essential source of self-awareness and personal identity and contributes to the regulation of behavior and the maintenance of physical and mental health. The aesthetic appearance of the breast is indeed of utmost importance to a woman’s sense of femininity, self-esteem, and self-confidence, and a woman’s erotic sensitivity [9]. Accordingly, inner perceptions are strongly related to body image (BI).

BI is defined as an individual’s subjective evaluation of oneself; it is a mental representation that involves thoughts, feeling, and perceptions related to body and appearance on cognitive, behavioral, and affective levels. Specially, BI constructs include three aspect: (a) body reality: the body as it actually exists; (b) body ideal: the image in one’s mind of how they would like the body to appear and behave, including norms of body contour, body space, and boundaries; and (c) body presentation: how the body is presented to the external environment [10].

According to this concept, BI can be strongly associated with body investment and body image evaluation. The first factor refers to the importance attributed to their overall appearance; the second aspect relates to an individual’s level of body satisfaction or dissatisfaction, including evaluative attitudes.

Furthermore, emotions play a relevant role in BI evaluation.

Women with more depression, anxiety, and physical symptoms are more likely to report distress related to body image concerns after surgery.

Thus, an earlier identification of patients’ characteristics and needs will permit us to relate actions in a dynamic decision trajectory that can improve patient and clinician communication [11,12].

### 4.2. Breast Anthropometry and Quality of Life

The literature reports several attempts to make accurate estimates of breast morphology after surgery. A number of methodologies have been employed to assess breast volume, from the most basic, such as thermoplastic casts [13] or water displacement [14], to the most complex diagnostic procedures such as MRI, CT scan, or digital surface scan [15]. Still, nowadays, none of these procedures are included in a core set of outcomes or are part of a standard assessment. Some software such as BCCT core (version 3.1) [16,17] or BAT (version 0.9.4.1) [18] perform a basic assessment of breast morphology and correlate it to the judgement of an expert panel with the purpose to transform objective assessments into quantitative estimates of cosmetic results.

Although cosmetic results are intuitively associated with breast shape, in reality, these results are highly subjective and cannot be easily represented only by geometric quantifications.

Nowadays, a consolidated trend includes quality of life as one of the main endpoints of breast surgery. The use of the Breast Q questionnaire has become generalized [19,20,21], and several translations and cultural adaptations are now available [22,23].

We queried if basic linear measurements can be representative of women’s satisfaction, so that they could be used to benchmark surgical outcomes in a quick, cheap, repeatable, and reproducible way.

The results of this study confirmed our hypothesis and found a relevant association between nipple to sternal notch distance, body mass index, and areola to infra-mammary fold distance and breast related quality of life in a population of asymptomatic women (irrespective of age). This association is negative and highly statistically significant.

In the past, other studies correlated body anthropometry to Breast Q domains, creating baselines or investigating factors associated to pre-operative breast-related values of quality of life.

A retrospective analysis of prospectively collected data assessed 13,063 women candidates of post-mastectomy breast reconstruction who preoperatively completed the Breast Q questionnaire. The mean preoperative satisfaction with breasts score was 61.8 ± 21.5 and the median score was 58.0 (interquartile range, 48 to 70). Factors associated with significantly lower preoperative satisfaction included a history of psychiatric diagnosis, preoperative radiotherapy, marital status (married), and a higher body mass index [24].

The Breast Q questionnaire was also used on a large population of asymptomatic women to create normative values. Women with a body mass index of 30 kg/m^2^ or greater, and a cup size of D or greater, at an age younger than 40 years confirmed poorer scores [25]. High BMI score are reported in association with low pre-operative satisfaction in other studies, confirming that this value may impact breast satisfaction scores [26].

All these results are in keeping with what is reported in our series, showing a significant correlation between BMI and breast-related quality of life after introducing a correction for age (negative correlation with satisfaction with breasts, Spearman partial correlation coefficient, SPCCC = −0.31 *p* = 0.0002, and overall satisfaction, SPCC = −0.21 *p* =0.0124). However, this is not the only value we investigated.

Another study investigated the correlation between breast volume, body mass index, and breast rating, calculated using the Breast Size Rating Scale. The results confirmed that greater breast size dissatisfaction was significantly associated with lower self-esteem, greater body-dissatisfaction, and higher BMIs [27].

Therefore, it is clear that there is a correlation between BMI, breast volume, and women’s satisfaction with their breasts.

If the calculation of body mass index is very easy, on the contrary, breast volume estimation requires unpractical or uncomfortable evaluations or highly complex and expensive methodologies.

Historically, a very simple and reliable method for volume estimation was the thermoplastic cast, with a 6% error in final estimations and some clear discomfort to patients for a methodology that cannot be easily applied to everyday clinical practice [28].

Three-dimensional surface image scanning was tested on 29 patients by Killaars et al. [28], and the measures were correlated with estimates of breast volume in MRI with an excellent intra-class correlation (0.991) comparable to that of MRI. However, surface scanning may have several limitations, especially in patients with large and ptotic breasts whose lower pole is in contact with the chest wall surface, preventing reliable assessments of this area [4].

Recently, machine learning was used to predict breast volume from basic linear measurements. Despite the very promising results, still, the procedure looks rather complex, and its repeatability and reproducibility are required to be tested on a larger scale with less trained operators [29].

Nipple to sternal notch distance and areola–inframammary fold distance are well-known determinants of breast appearance. Historical studies on breast anthropometry describe these linear measurements to be highly correlated with volume and shape, with larger volumes associated with higher distances between relevant landmarks. Back in 1955, Penn described a few distances that indicated a cosmetic breast shape [30].

Some authors tried to find the relationship between distances and cosmetic appearance. Liu et al. gathered an expert panel made up of surgeons, cancer patients, and cosmetic surgery patients. They concluded that the ideal sternal notch to nipple distance was 21 to 21.5 cm, the ideal nipple to IMF distance was 8 cm, and the idea nipple to nipple distance was 21 cm. Body mass index was also associated with distances, revealing larger differences between lighter (BMI < 25 kg/m^2^) and heavier (BMI > 25 kg/m^2^) women. The authors concluded that ideal measurements in the heavier group were always slightly larger [31].

All these investigations prove an association between distances, volume, and BMI, that is a demonstrated predictor of breast-related quality of life [24,25,26,27].

Considering the difficulty of obtaining an easy and quick measurement of volume, we substituted it with three easy distances that were proven to have a significant association with quality of life in the following domains: “Satisfaction with Breast”: SN–N; A–IMF; N–N; “Sexual satisfaction”: SN–N. After corrections for age, still, SN–N and A–IMF (together with BMI) were negatively correlated to satisfaction with breasts with high statistical significance.

## 5. Conclusions

In other words, by the time we act to reduce relevant measures, we are likely to increase breast-related quality of life, so that shorter distances can be considered a benchmark for outcome optimization.

In clinical practice, anthropometry can also be used to inform the shared decision making process, identifying patients who are likely to benefit from therapeutic mammoplasties.

Linear distances and BMI can be considered good predictors of BRQoL. These may be monitored easily and quantitatively over time, in contrast from questionnaires that are usually administered once or twice after surgery.

Interestingly, scores in the four domains of the Breast Q questionnaire reported in this observation are rather low compared to other normative values [25], but not different from those described in another population of women coming from the same geographical area [32].

This confirms that PROMs are highly culturally dependent and the importance of having local baseline values to make reliable comparisons.

These results are limited by the small size of the sample (145 healthy women), which probably reduced the statistical significance of some domains and some measures. Another important limitation of this study is related to the lack of information on other factors that may affect quality of life. For instance, we did not assess co-morbidities, psychiatric disorders, or other psychosocial aspects that go along with increases in BMI.

A new study should be performed on surgical candidates including pre- and post-op interviews to see to what extended single measures taken pre-operatively may predict final values of satisfaction.

## Figures and Tables

**Table 1 healthcare-12-00268-t001:** Patients’ characteristics.

Variable (N = 145)	Mean	Std Dev	Minimum	Maximum
Age (years)	44.5	10.8	20.5	79.3
Height (cm)	161.83	5.48	149	175
Weight (kg)	63.14	11.56	40	124
BMI (kg/m^2^)	24.12	4.33	17.26	47.25
SN–N distance (left) (cm)	24.5	3.92	17	41
SN–N distance (right) (cm)	24.42	3.91	17	38
A–IMF distance (left) (cm)	7.24	2.37	3	13
A–IMF distance (right) (cm)	7.22	2.49	3	14
N–N Distance (cm)	21.72	2.81	17	32

SN: sternal notch; N: nipple; A: areola; IMF: infra-mammary fold.

**Table 2 healthcare-12-00268-t002:** Breast Q values.

Variable	Mean Value
Psycho-social well-being (N = 145)	58
Physical well-being (N = 145)	64
Sexual well-being (N = 139)	56
Satisfaction with breasts (N = 139)	48

**Table 3 healthcare-12-00268-t003:** Spearman correlations.

		Overall Quality of Life	Psycho-Social Well-Being	Physical Well-Being	Sexual Well-Being	Satisfaction with Breasts
Age	R_s_	−0.18678	−0.05547	−0.09030	−0.21258	−0.23741
	*p*-value	0.0245	0.5076	0.2801	0.012	0.0049
BMI	R_s_	−0.33348	−0.11674	−0.06904	−0.16708	−0.28923
	*p*-value	<0.0001	0.162	0.4093	0.0493	0.0006
SN–N distance	R_s_	−0.37034	−0.11948	−0.06748	−0.17854	−0.28986
	*p*-value	<0.0001	0.1523	0.42	0.0355	0.0005
A–IMF distance	R_s_	−0.31460	−0.02956	−0.03735	−0.11937	−0.18239
	*p*-value	0.0001	0.7241	0.6556	0.1616	0.0316
N–N distance	R_s_	−0.14281	−0.07059	−0.11458	0.04119	−0.07121
	*p*-value	0.0866	0.3989	0.17	0.6302	0.4049

R_s_: Spearman correlation coefficient; SN: sternal notch; N: nipple; A: areola; IMF: infra-mammary fold.

**Table 4 healthcare-12-00268-t004:** Spearman correlations after normalization for age.

		Overall Quality of Life	Psycho-Social Well-Being	Physical Well-Being	Sexual Well-Being	Satisfaction with Breasts
BMI	R_s_	−0.31374	−0.11118	−0.04116	−0.08459	−0.21243
	*p*-value	0.0002	0.1942	0.6317	0.3239	0.0124
SN–N distance	R_s_	−0.32168	−0.08844	−0.07231	−0.10466	−0.21912
	*p*-value	0.0001	0.3023	0.3993	0.2219	0.0098
A–IMF distance	R_s_	−0.27853	−0.00862	−0.05562	−0.07452	−0.13561
	*p*-value	0.0009	0.92	0.517	0.385	0.1128
N–N distance	R_s_	−0.10743	−0.08396	−0.08116	0.1077	−0.00474
	*p*-value	0.2098	0.3275	0.344	0.2086	0.956

R_s_: Spearman correlation coefficient; SN: sternal notch; N: nipple; A: areola; IMF: infra-mammary fold.

## Data Availability

Data supporting the results are available in a dedicated database and are available upon request.

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
