# Peer review of "Easy Anthropometric Measurements Are Representative of Baseline Values of Breast Q Values in Asymptomatic Women"

_healthcare, 2024, doi:10.3390/healthcare12020268_

Round 1

Reviewer 1 Report

Comments and Suggestions for Authors

Overall, this is an interesting study, with the potential for impact within the field of breast reconstruction or cosmetic surgery and finding ways to assess patient satisfaction with surgical outcomes. However, I do have a few comments regarding the background and methodologies used in this study discussed below.

The authors correctly state at the beginning of the paper that breasts have a complex three-dimensional shape. However, the author then goes on to only collect simple one-dimensional measure of the participants' breasts, which by virtue of their one-dimensional nature cannot describe features of 3D shape. As a mathematical concept, the shape of an object is the information that describes it when the effects of scale, location and rotation have been removed and are therefore scale-invariant. It has been shown previously that humans perceive differences in body shape between individuals by identifying scale-invariant external features, such as curvatures, proportions and lateral contours, none of which can be captured using linear, one-dimensional measures (distances, girths, etc.). The author justifies not obtaining more sophisticated measures of breast shape due to the need for complex technologies to obtain reliable results. However, there have been a number of studies which have developed low-cost 3D imaging devices (utilising depth sensors and other imaging technologies, such as the xbox kinect), which can obtain measures such as volume, curvature and principal components of shape variation. Therefore, in my view, the authors decision to only collect simple measures limits the analysis that could have been performed, and is only assessing relationships that exist between quality of life and simple measures which are not perceivable by individuals. 

I would ask that the authors include more thorough justification for not collecting measures of breasts which would enable the assessment of differences in breast shape which greater granularity and the limitations of the methodology which was used in this study.

Author Response

Many thanks for this comment.

We agree with considerations about substantial differences among linear measurements, curvatures, proportions and contours. This is a topic extensively investigated by the first author and the team of Fondazione G.Re.T.A.

(Catanuto G, Taher W, Rocco N, Catalano F, Allegra D, Milotta FLM, Stanco F, Gallo G, Nava MB. Breast Shape Analysis With Curvature Estimates and Principal Component Analysis for Cosmetic and Reconstructive Breast Surgery. Aesthet Surg J. 2019 Jan 17;39(2):164-173.

Catanuto G, Spano A, Pennati A, Riggio E, Farinella GM, Impoco G, Spoto S, Gallo G, Nava MB. Experimental methodology for digital breast shape analysis and objective surgical outcome evaluation. J Plast Reconstr Aesthet Surg. 2008;61(3):314-8. 

Human Breast Shape Analysis using PCA G. GalloG. GuarneraG. Catanuto Published in International Conference on… 2010 )

Most of these previous experiences have been conducted either with low cost devices or using clinical imaging not specifically designed for morphological appraisal. Therefore, surface acquisition is possible and easy, but still, until now, we do not have a standardized protocol for this assessment which is perceived as time consuming, requiring specific facilities, and sometimes unreliable.

This study was designed to provide estimates of breast-related quality of life at a glance (i.e. nearly without interfering with clinical routine) in order to inform the shared decision making process. Further measurements are highly recommended especially into an experimental setting.

Reviewer 2 Report

Comments and Suggestions for Authors

In the manuscript, "Easy anthropometric measurements are representative of baseline values of BREAST-Q values in asymptomatic women," by Catanuto et al. the authors demonstrate a correlation between age, BMI, S-N and N-IMF distances with decreasing satisfaction. The study looked a relatively large number of patients (145) over a rather wide patient population. Although the findings of the study are not surprising this appears to be the first publication to demonstrate this. As one could assume that age plays a large role in effecting these they found persistent findings when normalizing for this. Some questions I have for you are:  You mention in introduction (line 52-53) setting a morphological benchmark but never discuss what this is in the paper and how this can be achieved clinically. In the discussion you talk about breast volume but never calculate breast volume you just use linear measurements so I am not sure you can comment on breast volume and PRO. Could other factors be effecting result, co-morbidities, psychiatric diagnoses, other psychosocial aspects that go along with increase BMI. Most importantly you demonstrate these correlations but how does this change practice or be implemented clinically? Finally any thoughts on why do your baseline values differ from Mundy et al. values and does this affect the translatabilty of findings to other cohorts of people.

- You mention in introduction (line 52-53) setting a morphological benchmark but never discuss what this is in the paper and how this can be achieved clinically.

Author Response

Many thanks for these comments that will certainly improve the quality of our brief report.

In the manuscript, "Easy anthropometric measurements are representative of baseline values of BREAST-Q values in asymptomatic women," by Catanuto et al. the authors demonstrate a correlation between age, BMI, S-N and N-IMF distances with decreasing satisfaction. The study looked a relatively large number of patients (145) over a rather wide patient population. Although the findings of the study are not surprising this appears to be the first publication to demonstrate this. As one could assume that age plays a large role in effecting these they found persistent findings when normalizing for this.

Some questions I have for you are:  You mention in introduction (line 52-53) setting a morphological benchmark but never discuss what this is in the paper and how this can be achieved clinically.

The information provided in study proves that breast related quality of life is associated with some relevant measurements. By the time we act reducing these measurements, we are likely to increase patient’s satisfaction. In other words shorter distances are likely to be considered the benchmark for optimization of quality of life. We believe this is relevant information that can be used to inform the shared decision making process, and assess patients profiles preliminary without evaluation of complex geometrical properties (surface curvature; volume; deformation).

In the discussion you talk about breast volume but never calculate breast volume you just use linear measurements so I am not sure you can comment on breast volume and PRO.

The association of volume- and linear measurements is indirect and derived from other previous reports.

  1. BMI is associated to  QoL (ref 26-27)
  2. Linear measurement are highly correlated to volume (30)
  3. BMI is associated to linear measurements (32)

Syllogistically, in this study linear measurements can be representative of breast related quality of life and can be a quick surrogate of breast volume estimation.

Could other factors be effecting result, co-morbidities, psychiatric diagnoses, other psychosocial aspects that go along with increase BMI.

Unfortunately, this study was not designed to capture the impact of these variables. This is one of the most relevant limitations. We will mention it in the manuscript.

Most importantly you demonstrate these correlations but how does this change practice or be implemented clinically?

Data from this study suggest that simple measurements can anticipate values of breast related quality of life. Ideally, from now on clinicians should report at least BMI and SN distance in their clinical examination of every woman, so that they can estimate changes over time either in the setting of asymptomatic and symptomatic women.

For instance, a doctor who sees for the first time a woman in clinical practice, using the nipple to sternal notch distance and BMI will have some information on her QoL. These values can be recorded without complex estimations and can be followed over time to monitor QoL that normally can be assessedwith long questionnaires only once or twice after surgery.

Finally any thoughts on why do your baseline values differ from Mundy et al. values and does this affect the translatabilty of findings to other cohorts of people.

Thanks for this comment. We did observe variations with the Mundy study and also with other similar reports on baseline values of breast related quality of life. However, what reported in this study for asymptomatic women do not differ from the results of another study from the same author on a completely different population coming from the same area. In conclusion we believe that PROMs are highly culturally dependant and therefore local baseline values are mandatory.

Breast 2019 Aug:46:12-18. doi: 10.1016/j.breast.2019.04.002. Epub 2019 Apr 9. De-escalation of complexity in oncoplastic breast surgery: Case series from a specialized breast center

G Catanuto 1, A Khan 2, V Ursino 3, E Pietraforte 3, G Scandurra 3, C Ravalli 3, N Rocco 4, M B Nava 5, F Catalano 3

Affiliations expand PMID: 30999077 DOI: 10.1016/j.breast.2019.04.002